# ODEM-GAN: An Object Deformation Enhancement Model Based on Generative Adversarial Networks

Zeyang Zhang [1], Zhongcai Pei [1], Zhiyong Tang [1,*] and Fei Gu [2,*]

[1] School of Automation Science and Electrical Engineering, Beihang University, Beijing 100191, China; zzy89087@buaa.edu.cn (Z.Z.); peizc@buaa.edu.cn (Z.P.)
[2] School of Computer Science and Technology, Soochow University, Suzhou 215006, China
* Correspondence: zyt_76@buaa.edu.cn (Z.T.); gufei@suda.edu.cn (F.G.)

**Abstract:** Object detection has attracted great attention in recent years. Many experts and scholars have proposed efficient solutions to address object detection problems and achieve perfect performance. For example, coordinate-based anchor-free (CBAF) module was proposed recently to predict the category and the adjustments to the box of the object by its feature part and its contextual part features, which are based on feature maps divided by spatial coordinates. However, these methods do not work very well for some particular situations (e.g., small object detection, scale variation, deformations, etc.), and the accuracy of object detection still needs to be improved. In this paper, to address these problems, we proposed ODEM-GAN based on CBAF, which utilizes generative adversarial networks to implement the detection of a deformed object. Specifically, ODEM-GAN first generates the object deformation features and then uses these features to enhance the learning ability of CBFA for improving the robustness of the detection. We also conducted extensive experiments to validate the effectiveness of ODEM-GAN in the simulation of a parachute opening process. The experimental results demonstrate that, with the assistance of ODEM-GAN, the AP score of CBAF for parachute detection is 88.4%, thereby the accuracy of detecting the deformed object by CBAF significantly increases.

**Keywords:** deformation enhancement; generative adversarial network (GAN); coordinate-based anchor-free (CBAF); parachute detection

## 1. Introduction

With large-scale datasets [1,2], object detection based on convolutional neural networks (CNNs) has achieved impressive performance. These CNN-based methods are not only improving bounding box object detection [3–5], but also making progress on local correspondence [6,7]. However, for some particular situations, such as small object detection, scale variation, and deformations, the traditional CNN-based methods do not perform very well. To address this problem, some scholars have proposed to build multi-level feature pyramids and expand datasets for improving the detection accuracy. The former is from the perspective of knowledge modeling, and the latter is from a data-driven perspective. Most state-of-the-art approaches are based on preset anchor-boxes [8–13] or use the feature pyramid network (FPN) [5,14–17] module to solve small object detection and scale variation.

Although these methods achieve good performance, there are still some inherent limitations. First, these methods tile a large number of artificially preset anchor boxes, and then predict the category of these anchor boxes. To obtain the high intersection over union (IOU) rate between anchor boxes and ground-truth boxes, they refine these anchor boxes one or more times, which will produce too many negative samples and considerable computational redundancy during the prediction process. Second, the anchor boxes have predefined scales and aspect ratios, which belong to discrete space. However, the ground-truth boxes belong to continuous space, so in object detection, there will be anchor boxes

and ground-truth boxes that are difficult to accurately align, which will lead to greater loss regression. For small object detection and scale variation, using feature pyramids is the principal method. Small objects are mapped to high-resolution feature maps (the low-level feature maps of the feature pyramid) [17] as they have fine-grained appearance information [18,19]. The feature pyramid is the basic module of object detection at different scales. In addition, the object deformation should also be taken into consideration during detection. A common method to solve deformation is to use a data-driven learning strategy, that is, to collect as many images in different states as possible, and use it to train the detector network to improve the performance of the detector [20,21]. However, it is impossible to collect the state of the object all the time. Moreover, the cost required to collect and label a large amount of data is also inconceivable.

To tackle these limitations, Tang et al. proposed a coordinate-based anchor-free (CBAF) module, which can effectively detect rigid objects of small and scale variation [22]. However, it does not work well with deformation objects such as parachutes. The reason is that the basic principle of the object detection algorithm is to calculate the similarity of the feature on the search image. When the similarity is greater than the pre-set threshold, it is considered the same object. If the parachute is just thrown as the object, its violent shape change will cause the confidence score during the detection process to drop sharply, resulting in detection failure (the detection object is lost multiple times during the experiment). If the fully opened parachute is used as the target, although the parachute will no longer deform drastically afterwards, the algorithm can stably detect the target, but when the parachute is just thrown from the aircraft, the parachute is not fully opened, and the detection algorithm cannot detect the parachute. As shown in Figure 1a–d, the movement process of the parachute is divided into four stages: (1) The free-fall stage of the airdropped item leaving the aircraft (Figure 1a); (2) The stage of pulling out the main parachute from the umbrella bag (Figure 1b); (3) The stage of inflating the main parachute (Figure 1c); and (4) The stage of stable descent of the main parachute (Figure 1d). If we choose the parachute in the stable descent phase as the training sample, it is difficult for the detector to detect the parachute in the first three phases. Because the parachute is gradually opened, its violent shape change will cause the confidence score during the detection process to drop sharply, resulting in detection failure (the detection object is lost multiple times during the experiment). If a severely deformed object is detected in a continuous image sequence, the boundary box will be inaccurately positioned and severely jitter. Figure 2 shows that the jitter of the bounding box is extremely unfavorable for the subsequent analysis of parachute shape and aerodynamic layout.

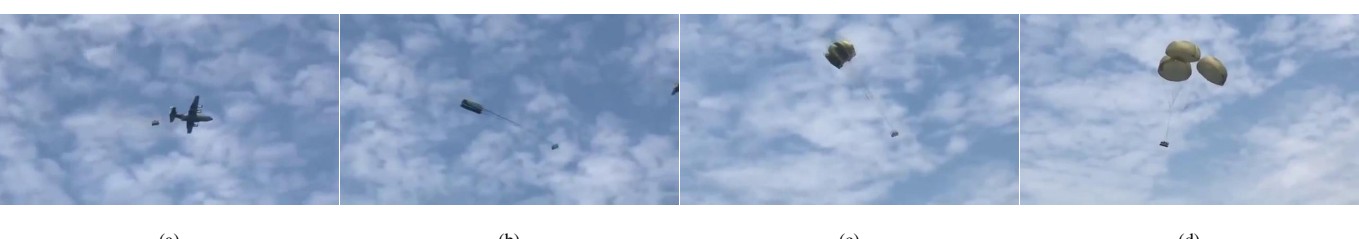

　　　　　　(a)　　　　　　　　　　　　　　　(b)　　　　　　　　　　　　　　　(c)　　　　　　　　　　　　　　　(d)

**Figure 1.** The process of parachute movement.

In order to address these issues, we propose ODEM-GAN, an object deformation enhancement model based on a generative adversarial network (GAN) for object detection. ODEM-GAN has two branches: one is the CBAF branch and the other is the GAN branch. The CBAF branch is used for small- and scale variation object detection, which divides the feature map into several patches by spatial coordinates, and each of these patches has a size of $m \times m$. These patches are used as the input of the contextual merging layer, and the output of the contextual merging layer, regarded as contextual feature-patches, is used for classification and regression, as shown in Figure 3. The GAN branch uses contextual feature

patches to generate deformation features for enhancing the performance of classification and regression networks.

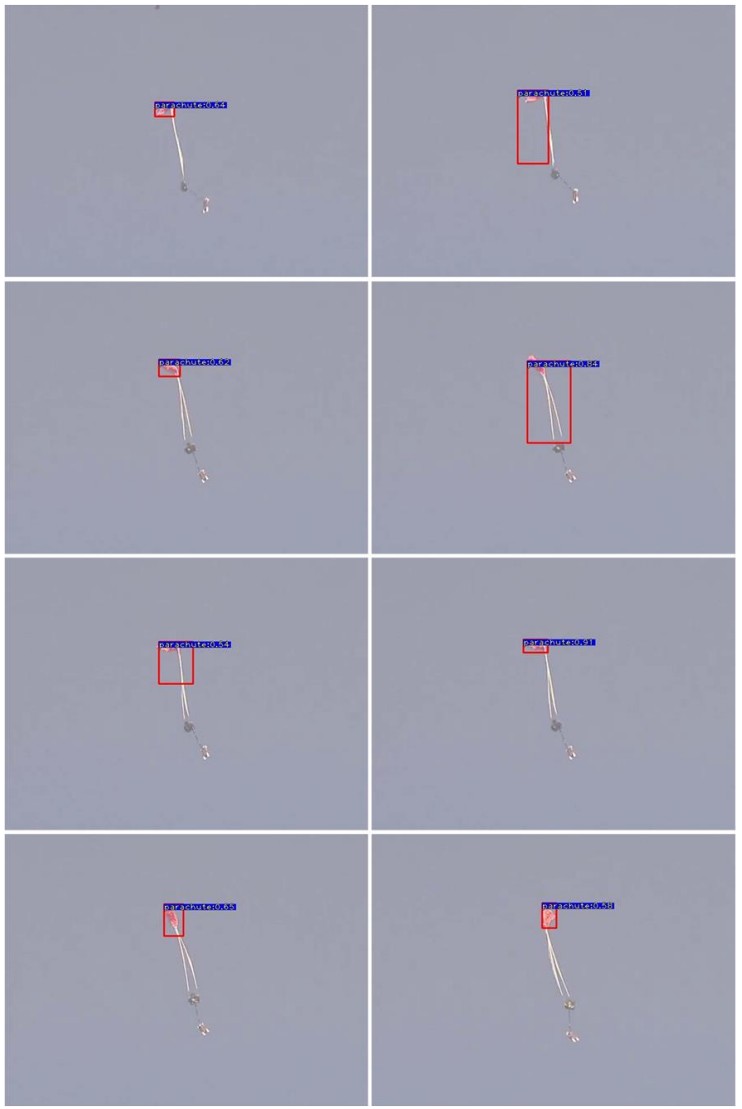

**Figure 2.** The detection of the continuous image sequence (timing: from left to right, top to bottom).

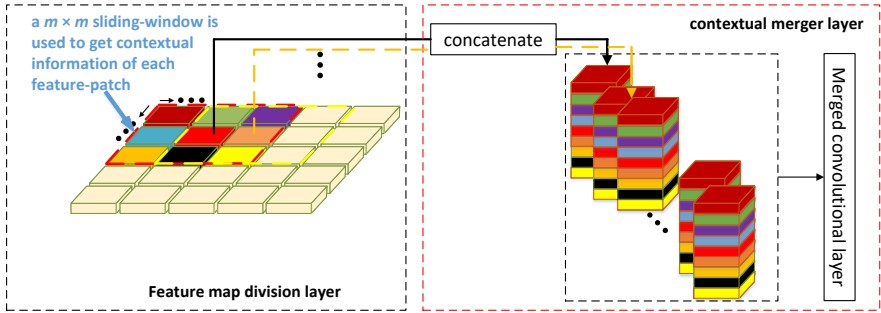

**Figure 3.** General concept of the proposed CBAF module.

We highlight our main contributions as follows:

- We propose ODEM-GAN, an effective object deformation enhancement model based on GAN, which focuses on the irrelevance of object deformation and scale variability to improve the prediction of hard sample probabilities and coordinates.

- In ODEM-GAN, an online deformable sample creation module and an anchor-free box prediction layer are developed. The online deformable sample creation module is to deform and augment the dataset. The anchor-free box prediction layer not only avoids accurately aligning discrete spatial anchor boxes and ground-truth boxes but also reduces the amount of calculation.
- We propose the positioning accuracy loss which reduces the bounding box's inaccurate positioning and jitter caused by the detection of severely deformed objects in a continuous image sequence. The hard deformed samples generated by GAN can improve the regression accuracy of the object detector.
- The experimental results on the airdrop dataset show that our ODEM module has obvious advantages in object detection with severe deformation. Moreover, the experimental results on MS-COCO show that the ODEM module also has advantages for small target detection, while improving the baseline of small object detection. In addition, the method also provides a new idea for object detection to replace object tracking algorithms.

The rest of this paper is organized as follows. Section 2 presents a review of related works. The main procedures and ODEM-GAN are studied in Section 3. Comparison performance experiments are given in Section 4. Finally, the conclusion is drawn in Section 5.

## 2. Related Works

Most object detection approaches are classified as one-stage methods and two-stage methods. The two-stage approach for object detection includes two steps. First, the extraction of multiple regions of interest (RoI) using the region proposal network. Second, these ROIs are classified and refined using the classification and regression module, respectively. Unlike the two-stage method, the one-stage method predicts the item immediately using a single neural network structure. The one-stage approach has a faster prediction speed than the two-stage method, but its accuracy is not as excellent. Among them, YOLO, RetinaNet, CornerNet, etc. are the most representative methods, and the contributions of these methods are described in detail in the next paragraph. Feature pyramids, also known as multi-level feature towers, are progressively becoming a frequent structure for object detection, since they handle object scale variances and produce considerable results. SSD [11] first predicts class scores and bounding boxes at different resolutions of feature maps so that objects of different sizes can be processed. FPN [14] and DSSD [5] proposed a top–down architecture with lateral connections which enhance low-level features with high-level semantic feature maps at all scales. In order to better adapt to the scale changes of anchors and to solve the limitations of anchor boxes, the anchor-free approach has attracted much attention in recent years. Among them, YOLO and CornerNet left a deep impression. CornerNet solved object detection as the key point detection problem, wherein the prediction box was obtained by detecting two key points in the upper left corner and the lower right corner.

### 2.1. Object Detection

Recent object detection approaches are either CNN approaches or transformer approaches. Using a transformer to complete visual tasks has become a new research direction to reduce the complexity of structures and explore scalability. The transformer performs better than CNNs in various computer vision tasks. Among them, the vision transformer (VIT) [23] is the pioneering work of the transformer in the field of computer vision. VIT divides the image into fixed-size patches, adds position embedding to the patch, preserves space/location information in the global scope through different strategies, and then inputs the linear projections of these patches together with their position codes into the transformer. Although these transformer-based approaches achieve good performance, they need to consume many computing resources and training data during the training process.

In the case of a small dataset, the performance of the CNN approach is equally impressive. The CNN approaches are divided into a two-stage approach and a one-stage approach. The one-stage method is currently a popular research direction. RetinaNet [13]

mainly makes the one-stage network accuracy better than the two-stage network, and the focal loss is proposed to solve the problem of positive and negative sample imbalance. FPN [5] proposes a top–down architecture with lateral connections, which enhances low-level features with high-level semantic feature maps at all scales. DetNet [14] designs a new backbone network to maintain a higher spatial resolution in the upper layers to improve the accuracy of bounding box regression. However, they all use pre-set anchor boxes to regress the bounding boxes. With the recent development of the idea of anchor-free detection, YOLO [12,17,24] is impressive in balance speed and accuracy. DenseBox [25] first proposes a fully convolutional framework that directly predicted bounding boxes. CornerNet [26] solves object detection as the key point detection problem, and the prediction box is obtained by detecting two key points in the upper left corner and the lower right corner. Tang et al. proposed the CBAF model [22], which uses contextual information to effectively detect the parachute. Inspired by their work, we propose ODEM-GAN in this paper to improve the performance of CBAF in terms of the accuracy.

### 2.2. Generative Adversarial Networks

GAN is a powerful generative model based on deep learning. Since it was proposed by I. Goodfellow et al. in 2014 [27], GAN has been successfully applied in many fields, such as computer vision and natural language processing. In recent years, GANs are also unitized to tackle object detection problems. The main approaches are divided into two categories. The first is to create a spatial deformation by occluding certain regions on a feature map or manipulating feature maps to generate an adversary network of hard samples [28,29], as shown in Figure 4. The key idea is to create adversarial samples in the convolutional feature space instead of directly generating pixel-level data. The other is to generate super-resolution feature maps to reduce the difference in representation between the small and large objects for improving small object detection performance. Specifically, the generator learns to transform small object representations into super the resolution representations that are sufficiently similar to real large objects to deceive against the discriminator. Perceptual GAN [30], the model discriminator, and the generator are confronted to identify the generated representation and impose conditional requirements on the generator; the generated small object representation must be conducive to the detection of the target. The discriminator of the multitask generative adversarial network (MTGAN) [31] is a multitask network that uses true/false scores, object category scores, and bounding box regressors to describe each super-resolution image patch. Compared with these methods of directly generating pixel-level images and directly expanding the dataset, it is easier to generate samples in the convolutional feature space.

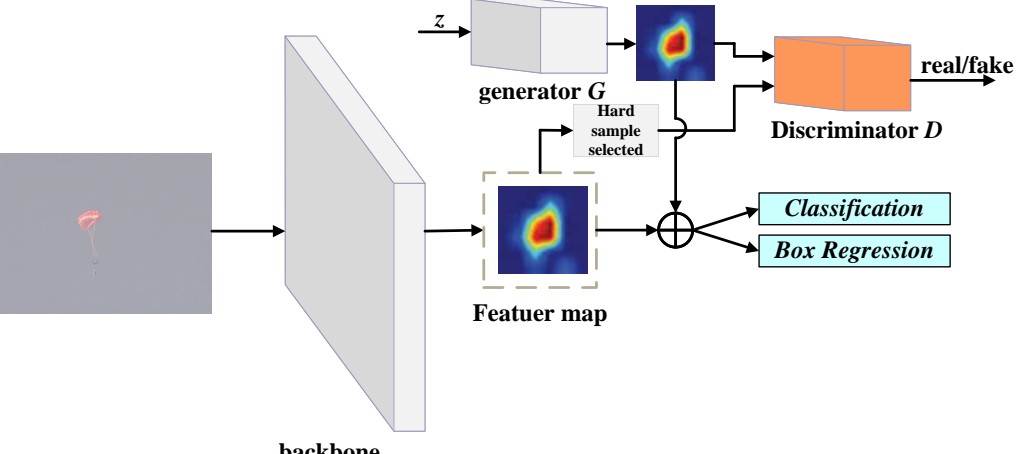

**Figure 4.** Overview of GANs-based object detection.

Therefore, inspired by the methods proposed by Wang [29] and Li [30], we used GAN networks to generate deformation samples. Unlike their methods, our method does not use masks or generate similar training samples. We generate deformation samples to achieve the detection of objects with dramatic deformation. For the problem of the accuracy of the one-stage method not being as good as that of the two-stage method, we used focal loss [13] to solve it. The Vit [23] and YOLO [12] algorithms achieve the establishment of object contextual relations by dividing the feature map, thus improving the accuracy of small target detection. Vit is a transformer-based detection network as it requires a large amount of training data, while YOLO is not as accurate as the two-stage network, although its prediction is fast. In this paper, we used object contextual relations to improve the accuracy of small object detection, then propose the ODEM-GAN module.

## 3. Approach

In this section, we describe the design process of the ODEM-GAN module, and show how to apply it to the backbone. We instantiate our ODEM-GAN module by using the state-of-the-art RetinaNet as the baseline. The design of ODEM-GAN is described from the following aspects: (1) how to choose hard samples for GAN; (2) how to design the GAN generator; (3) how to design the GAN discriminator; and (4) how to train and test the ODEM module and the branch of GAN.

### 3.1. Overview

To facilitate the demonstration, we use the $512 \times 512$ resolution of the image as input, and only $PF_3$ (the third of feature pyramid) of CBAF [22] is introduced; the other levels are the same. The resolution of $PF_3$ is $64 \times 64$ and the channel is $c = 256$. Therefore, the shape of $PF_3$ is $[b, 64, 64, 256]$ ($b$ is the batch size). As shown in the black-dashed region of Figure 3, the feature map division layer takes the feature map of $PF_3$ as input and divides the feature map of $PF_3$ into $N_{fp}$ feature patches of size $n * n$ according to the space coordinates. We set $n = 8$. Therefore, the shape of a feature patch is $[b, n, n, c] = [b, 8, 8, 256]$, the number of feature patches for the $PF_3$ level is $N_{fp} = \frac{h_3}{n} \times \frac{w_3}{n} = \frac{64}{8} \times \frac{64}{8} = 64$, and the output of the feature map division layer is $[b, N_{fp}, n, n, c] = [b, 64, 8, 8, 256]$. After obtaining the output of the feature map division layer, the feature patches are used as input for the contextual merger layer to concatenate with the surrounding feature patches, as shown in the red-dashed region of Figure 3. We used a $m * m = 9$ (we set $m = 3$) sliding window to obtain the contextual information of each feature patch; second, we concatenated the $m * m$ feature patches obtained in the sliding window along their channel dimensions. The shape of the merged feature patch is $[b, n, n, c \times m * m]$; finally, the merged feature pieces are passed to a $3 \times 3$ convolutional layer with $c_f = 512$ filters. The output shape of the contextual merger layer is $[b \times N_{fp}, n, n, c_f] = [b \times 64, 8, 8, 512]$.

A GAN is composed of a generator network and a discriminator network, and its structural diagram is shown in Figure 5 [27]. The generator network (abbreviated as $G$) uses a given noise vector $z$ (generally subject to uniform distribution or normal distribution) to generate data. The discriminator network (abbreviated as $D$) is used to discriminate whether it is generated data or real data. The former tries to generate more real data, while the latter tries to perfectly distinguish between real data and generated data. The two learn from each other in the adversarial process and make progress together until the data generated by $G$ become perfect so that $D$ cannot distinguish. We can consider $D$ as a binary classifier, so the basic model of GAN can be represented by cross entropy. The formula is shown in Equation (1):

$$\min_{G} \max_{D} V(D, G) = E_{x \sim p_{data}(x)}[logD(x)] + E_{z \sim p_z(z)}[log(1 - D(G(z)))], \quad (1)$$

where $x$ is the real data, $logD(x)$ represents the probability of the discriminator network on the real data, $G(z)$ represents the data generated by $G$, and $log(1 - D(G(z)))$ represents the

probability of *D* on the generated data. Through the max-min game, *G* and *D* are trained alternately until *D* cannot distinguish between the real and fake data.

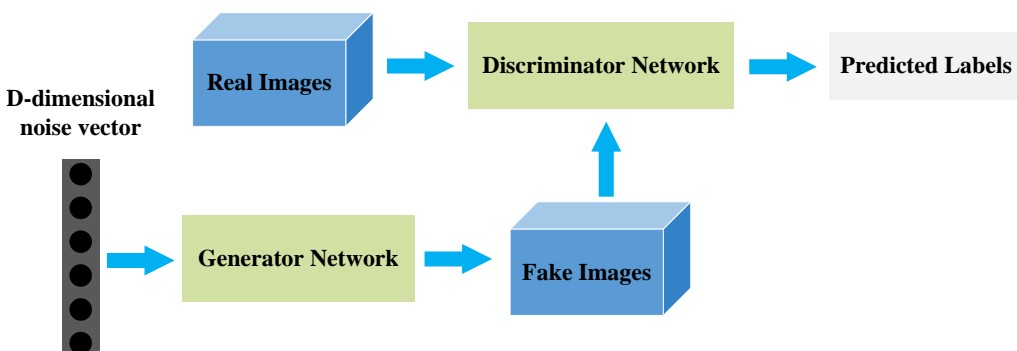

**Figure 5.** The process of generating an adversary network.

Similar to traditional GAN, we propose that ODEM-GAN also consists of *G* and *D*, which compete by adversarial loss and alternating training. However, the difference is that this paper adopts conditional GAN (conditional GAN, referred to as C-GAN); that is, the input of *G* not only has a noise vector *z* but also a given condition. Adding conditions to *G* cannot only generate samples under the conditions (specify the desired object that hard deformation samples are generated) but also reduce the training difficulty of GAN. For the overall structure of ODEM-GAN, this paper adopts a structure similar to that of BE-GAN [32]. The structural feature of BE-GAN is that *D* adopts the structure of the autoencoder, which is different from the data distribution of the learning samples of other GAN models. BE-GAN learns the distribution of the loss of the autoencoder through the Wasserstein adversarial loss.

As shown in the blue-dashed region of Figure 6, the ODEM-GAN module includes a CBAF module, a GAN branch, and a classification and regression module for each level of the feature pyramid.

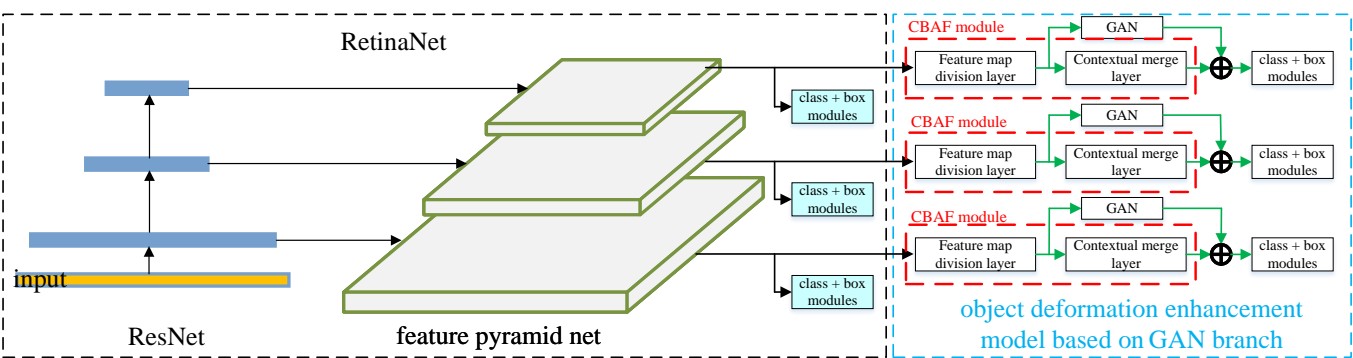

**Figure 6.** Overview of network architecture.

*3.2. Network Architecture of the ODEM-GAN Module*

3.2.1. ODEM-GAN Module

Figure 7 shows the GAN branch in the red-dashed box includes an online deformable sample creation module, a generator *G*, and a discriminator *D*. Among them, the online deformable sample creation module is used to choose and create the deformation samples that are difficult to predict by the classification and regression module (which refers to the hard sample feature of the deformation) and uses the deformation samples as the input of *D*. *G* is used to generate diverse deformation feature patches and add them to the merged feature patches with context information, which is used as the input of the classification and regression module so that the detector can learn more features of the deformed samples and improve the purpose of the detector to stably recognize the deformed samples. *D* is

used to compete with *G* so that *G* can generate a variety of deformed samples. Note that the GAN branch only participates in training. Note that GAN can only generate one certain category of object.

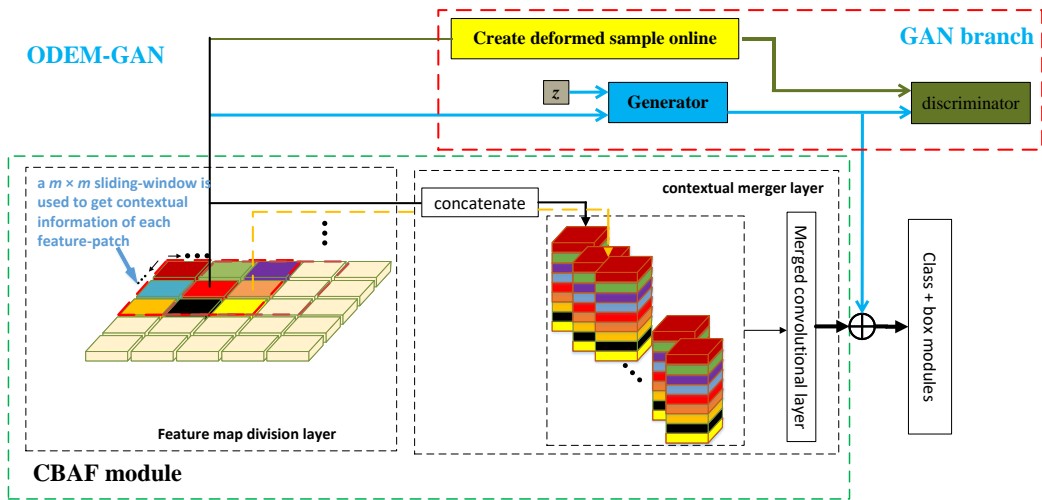

**Figure 7.** ODEM-GAN module includes a CBAF module (in the green-dashed box), a GAN branch (in the red-dashed box).

**Online deformable sample creation module.** To obtain more hard samples, the most direct method is the horizontal and vertical flipping of existing hard samples. Therefore, this module proposes an online creation module for deformed samples, as shown in Figure 8. In Section 3.1, we obtain the current feature patches and contextual feature patches and the flip operation on the current feature patches and their contextual feature patches to obtain the new deformation samples. Specifically, if the sliding window contains the feature patches of hard samples, the online deformable sample creation module flips contextual feature patches horizontally, vertically, on the left diagonal and right diagonal in the sliding window (see the red-dashed box on the feature map in Figure 8). However, the current feature patches in the sliding window do not contain the feature patches of hard samples, the contextual feature patches are filled with 0, and the flip operation is not performed.

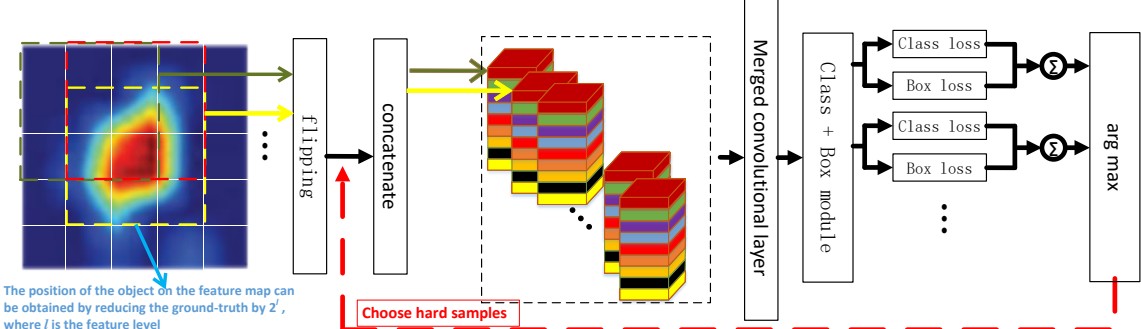

**Figure 8.** Online deformable sample creation module. Each sample passed through the CBAF module to compute the focal loss and smooth $L_1$ loss. Then, the level with the largest sum of the two losses is selected as the supervision signal for hard samples.

The flip operation obtains four new hard sample deformation features (we call the current feature patch and its contextual feature patches features) which are used as the input of the contextual merger layer to obtain new merged feature patches. Then, the new merged feature patches are used as the input of the classification and regression modules, and the sum of their classification loss and regression loss is calculated. What the detector needs is to make it difficult to classify and regress samples, and the purpose of improving robustness

and detection accuracy by learning these hard samples. The greater the loss of the sample, the more difficult it is for the detector to predict it. Therefore, the merged feature patches corresponding to the maximum loss are selected as the hard samples. Through the above steps, we can complete the online creation of deformation samples.

**The design of generator $G$.** ODEM-GAN is built on each level of the pyramid feature, and the parameters of each GAN branch are not shared. For ease of presentation, only the ODEM-GAN module of $PF_3$ is introduced, and the others are the same. As shown in Figure 9, the input of $G$ is the noise vector $z$ and the current feature patch, where $z$ can increase the diversity of the generated samples, and the current feature patch is used as the condition of the generator, which can ensure that the expected sample is generated or can reduce the difficulty of training. Specifically, $z$ is a 64-dimensional vector that is a uniform distribution of $[-1, 1]$. We take the average of the feature patches along the channel dimension; then, the shape of the feature patches on $PF_3$ will change from $[n, n, c] = [8, 8, 256]$ to $[8, 8, 1]$ and flatten it into a 64-dimensional vector. Finally, the 64-dimensional vector is concatenated with $z$ to obtain a new 128-dimensional vector as the input of the generator. The 128-dimensional vector achieves a shape of feature patches $[n, n, c_f] = [8, 8, 512]$ by upsampling (the upsampling strategy uses nearest neighbor interpolation) and three convolution modules.

As shown in Figure 9, the number of channels of the convolutional layer is the same as the multiple of the resolution increase. There are two upsamplings, and $n_c = \frac{c_f}{2^2} = 128$ in Figure 9. The parameters of generator $G$ are shown in Table 1.

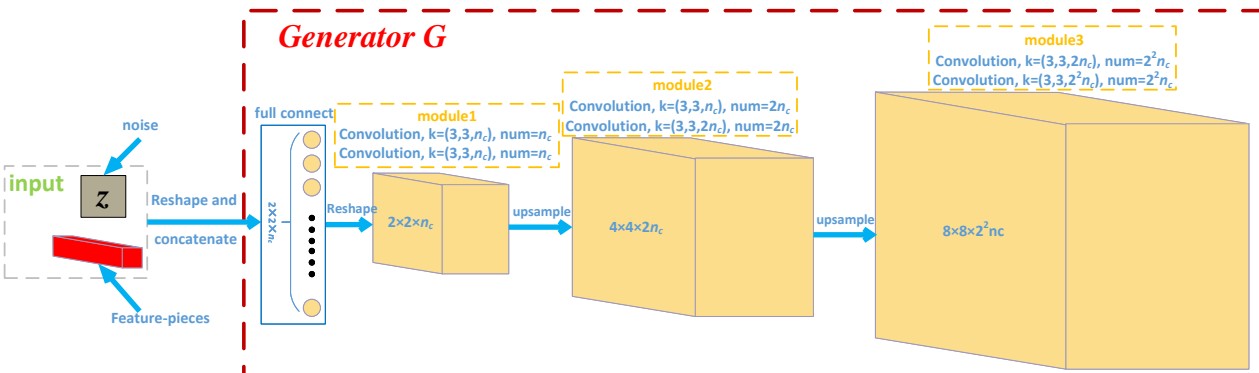

**Figure 9.** The structure of generator $G$.

**Table 1.** The parameters of generator $G$.

| Name | | Output Size (Height, Width, Channel) | Act. | Layer (Filter_h, Filter_w, Num, Stride) |
|---|---|---|---|---|
| feature-patch + $z$ | | (128, -, -) | - | - |
| full connect layer | | ($2 \times 2 \times n_c$, -, -) | - | - |
| module1 | conv1 | ($2, 2, n_c$) | Relu | ($3, 3, n_c, 1$) |
| | conv2 | ($2, 2, n_c$) | Relu | ($3, 3, n_c, 1$) |
| module2 | conv1 | ($4, 4, 2n_c$) | Relu | ($3, 3, 2n_c, 1$) |
| | conv2 | ($4, 4, 2n_c$) | Relu | ($3, 3, 2n_c, 1$) |
| module3 | conv1 | ($8, 8, 2^2 n_c$) | Relu | ($3, 3, 2^2 n_c, 1$) |
| | conv2 | ($8, 8, 2^2 n_c$) | - | ($3, 3, 2^2 n_c, 1$) |

**The design of discriminator $D$.** As shown in Figure 10, the discriminator adopts an autoencoder structure. It consists of two parts: an encoder and a decoder. The encoder encodes the input (the samples created by the generator and online deformable sample creation module) into a set of multidimensional feature vectors, while the function of the decoder is the opposite of the encoder. This restores the multidimensional feature

vectors generated by the encoder to the shape of the original input. Therefore, the output dimension of the autoencoder is the same as its input dimension. The decoding structure used in this article is the same as the generator $G$. The encoder is the inverse process of the decoder. The parameters of the discriminator $D$ are shown in Table 2.

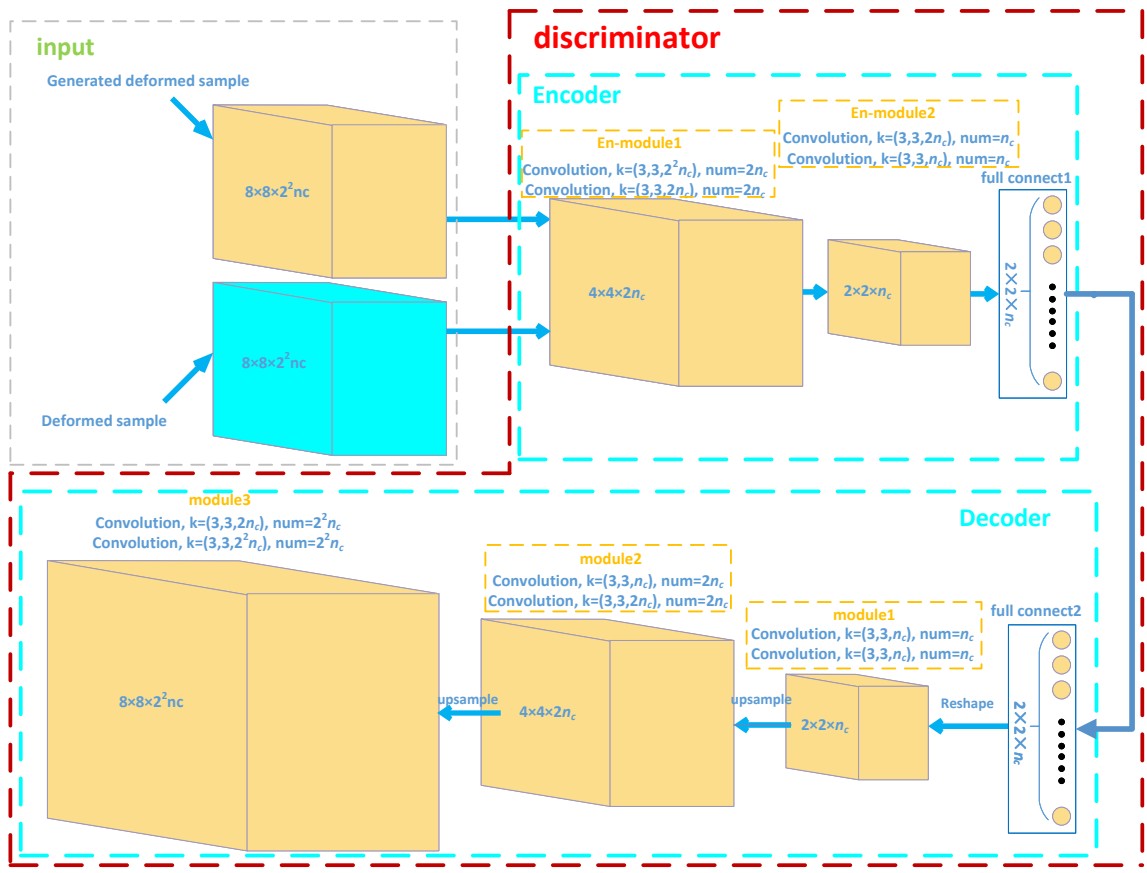

**Figure 10.** The structure of discriminator $D$.

**Table 2.** The parameters of the discriminator $D$.

| Name | | Output Size (Height, Width, Channel) | Act. | Layer (Filter_h, Filter_w, Num, Stride) |
|---|---|---|---|---|
| $x^s / x^r$ | | $(8, 8, 22n_c)$ | \multicolumn{2}{c}{Provided by generator and online deformable sample creation module} |
| en-module1 | conv1 | $(8, 8, 2n_c)$ | Relu | $(3, 3, 2n_c, 1)$ |
| | conv2 | $(4, 4, 2n_c)$ | Relu | $(3, 3, 2n_c, 2)$ |
| en-module2 | conv1 | $(4, 4, n_c)$ | Relu | $(3, 3, n_c, 1)$ |
| | conv2 | $(2, 2, n_c)$ | Relu | $(3, 3, n_c, 2)$ |
| full connect layer1 | | $(2 \times 2 \times n_c, \text{-}, \text{-})$ | - | - |
| full connect layer1 | | $(2 \times 2 \times n_c, \text{-}, \text{-})$ | - | - |
| | conv1 | $(2, 2, n_c)$ | Relu | $(3, 3, n_c, 1)$ |
| | conv2 | $(2, 2, n_c)$ | Relu | $(3, 3, n_c, 1)$ |
| | conv1 | $(4, 4, 2n_c)$ | Relu | $(3, 3, 2n_c, 1)$ |
| | conv2 | $(4, 4, 2n_c)$ | Relu | $(3, 3, 2n_c, 1)$ |
| | conv1 | $(8, 8, 2^2 n_c)$ | Relu | $(3, 3, 2^2 n_c, 1)$ |
| | conv2 | $(8, 8, 2^2 n_c)$ | - | $(3, 3, 2^2 n_c, 1)$ |

3.2.2. Loss of the GAN Branch

Since $L2$ loss will lead to blurred sample generation, $L1$ loss is used for the autoencoder, as shown in Equation (2):

$$L(x) = \|x - D(x)\|_1,\qquad(2)$$

where $L(x)$ is the loss of the autoencoder, $D(x)$ is the output of the autoencoder, and $x$ is the input of the autoencoder. The loss of generator $G$ is shown in Equation (3):

$$L_G = L(x^s),\qquad(3)$$

where $x^s$ is the sample generated by generator $G$. The loss of the discriminator is shown in Equation (4):

$$L_D = L(x^r) - L(x^s),\qquad(4)$$

where $x^r$ are samples created by the online deformable sample creation module. To enhance the diversity of the samples generated, a coefficient of balance $k_t$ balances the generator and discriminator in each alternate training. The update of $k_t$ is shown in Equation (5):

$$k_{t+1} = k_t + \zeta(\eta L(x^r) - L(x^s)),\qquad(5)$$

where $\zeta$ is the learning rate of $k_t$, and the parameter $\eta$ can control the proportion of sample diversity. In this article, we set $\zeta = 0.01$ and $\eta = 0.5$. After adding $k_t$, the loss function of the discriminator $L_D$ becomes the following Equation (6):

$$L_D = L(x^r) - k_t L(x^s).\qquad(6)$$

Although generator $G$ can generate sufficiently realistic hard samples by minimizing Equations (2) and (5), to make the generated hard samples help improve the regression accuracy of the detector, we define that positioning accuracy loss is introduced here. The positioning accuracy loss $L_{lp}$ is as shown in (7):

$$L_{lp} = \begin{cases} |t^*_{i,j} - t^g_{i,j}|^2 * 0.5 * 0.9, |t^*_{i,j} - t^g_{i,j}| < 1/9 \\ |t^*_{i,j} - t^g_{i,j}| - \dfrac{0.5}{0.9}, else. \end{cases}\qquad(7)$$

The hard sample generated by generator $G$ is taken as the input of the detector regression module, and after the forward propagation of the regression module, the 4-dimensional offset $t^g_{i,j} = [t^g_{t_{i,j}}, t^g_{l_{i,j}}, t^g_{b_{i,j}}, t^g_{r_{i,j}}]$ of the object bounding box will be output. Then, the smooth $L1$ loss of $t^g_{i,j} = [t^g_{t_{i,j}}, t^g_{l_{i,j}}, t^g_{b_{i,j}}, t^g_{r_{i,j}}]$ and the 4-dimensional offset $t^*_{i,j}$ of the ground-truth box are calculated.

The purpose of the positioning accuracy loss is to make the hard samples generated by the generator improve the regression accuracy of the object detector. Therefore, $L_{lp}$ is added to the loss of generator $G$. The loss function of generator $L_G$ becomes the following Equation (8):

$$L_G = L(x^s) + \lambda_{lp} L_{lp},\qquad(8)$$

where $\lambda_{lp}$ can control the weight of $L(x^s)$ and $L_{lp}$. In this article, we set $\lambda_{lp} = 0.1$. By minimizing Equations (6) and (8) and alternately training the discriminator $D$ and generator $G$, the generator can generate sufficiently realistic hard samples.

*3.3. Ground Truth and Loss of the Detector*

Since the GAN branch does not participate in the backpropagation of the detector, nor does it participate in inference, the loss of the GAN branch is not included in the loss of the detector.

### 3.3.1. Classification Modules

The ODEM-GAN module predicts objects based on the anchor-free module. This approach is similar to labeling for image segmentation. Pixels inside the bounding box are regarded as positive samples, pixels outside the bounding box are regarded as negative samples, and these pixels are filled with 0. The classification output is a feature map of $K$ channels, and each channel represents a class rather than an $N \times K$ dimensional probability ($N$ is the number of anchor boxes). The classification module consists of four $3 \times 3$ convolutional layers. Each convolutional layer has 256 filters, and ReLU is added after each convolutional layer. The output layer consists of a $3 \times 3$ convolutional layer with $K$ filters and sigmoid activation functions. We used focal loss [13] with hyperparameters $\alpha_t = 0.25$ and $\gamma = 2:0$ as the total classification loss for the ODEM-GAN module, as shown in Equation (9):

$$L_{cls} = -\alpha_t (1 - p_t)^\gamma log(p_t),\tag{9}$$

where $p_t = \begin{cases} p, y = 1 \\ 1 - p, else \end{cases}$, and $p$ is the output of the classification module.

### 3.3.2. Regression Modules

The structure of the regression module is the same as that of the classification module. The difference is that the output layer is a $3 \times 3$ convolutional layer with four filters. The regression module outputs four offset maps of the ground truth and the bounding box. The distance from each pixel to the top, left, bottom, and right of the bounding box is represented by a 4-dimensional vector $t_{i,j}^* = [t_{t_{i,j}}^*, t_{l_{i,j}}^*, t_{b_{i,j}}^*, t_{r_{i,j}}^*]$. The regression loss is the same as RetinaNet, using smooth $L_1$ loss, as shown in Equation (10).

$$L_{reg} = \begin{cases} \left| t_{i,j}^* - t_{i,j} \right|^2 * 0.5 * 0.9, \left| t_{i,j}^* - t_{i,j} \right| < 1/9 \\ \left| t_{i,j}^* - t_{i,j} \right| - \dfrac{0.5}{0.9}, else. \end{cases}\tag{10}$$

where $t_{i,j}$ is the 4-dimensional offset of the output of the bounding box.

Note that the parameters for classification are shared across each pyramid level, and the parameters for the regression module are also shared across each pyramid level. However, the parameters for classification and regression are independent. The architectures of the classification and regression modules are shown in Table 3.

**Table 3.** Only PF3 is introduced, similarly to in the other levels.

| Name | | Output Size (Height, Width, Channel) | Act. | Layer (Filter_h, Filter_w, Num, Stride, Padding) |
|---|---|---|---|---|
| classification module | class_conv1 | (8, 8, 256) × 64 | ReLU | (3, 3, 256, 1, 1) |
| | class_conv2 | (8, 8, 256) × 64 | ReLU | (3, 3, 256, 1, 1) |
| | class_conv3 | (8, 8, 256) × 64 | ReLU | (3, 3, 256, 1, 1) |
| | class_conv4 | (8, 8, 256) × 64 | ReLU | (3, 3, 256, 1, 1) |
| | class_output | (8, 8, K) × 64 | Sigmoid | (3, 3, K, 1, 1) |
| regression module | regress_conv1 | (8, 8, 256) × 64 | ReLU | (3, 3, 256, 1, 1) |
| | regress_conv2 | (8, 8, 256) × 64 | ReLU | (3, 3, 256, 1, 1) |
| | regress_conv3 | (8, 8, 256) × 64 | ReLU | (3, 3, 256, 1, 1) |
| | regress_conv4 | (8, 8, 256) × 64 | ReLU | (3, 3, 256, 1, 1) |
| | regress_output | (8, 8, K) × 64 | - | (3, 3, 4, 1, 1) |

The total loss of the ODEM-GAN is shown in Equation (11).

$$L_{ODEM-GAN} = \frac{1}{N_{cls}} \sum_{l=3}^{7} \sum_{i,j} L_{cls} + \frac{1}{N_{reg}} \sum_{l=3}^{7} \sum_{i,j} L_{reg},\tag{11}$$

where $N_{cls}$ is the total of the positive and negative samples and $N_{reg}$ is the total of positive samples.

*3.4. Joint Inference and Training*

Our ODEM-GAN module can work with the original RetinaNet, and all hyperparameters in training and inference are the same as those in RetinaNet. Note that the GAN branch of the ODEM-GAN module only participates in training and does not participate in inference.

**Model pretraining:** Experiments show that it is necessary to pretrain ODEM-GAN before using ODEM-GAN to improve the performance of the detector. The specific pretraining process is as follows:

- The training dataset: a dataset that only contains specific hard deformation objects that need to be detected;
- Use the trained CBAF object detection model to extract the features of specific hard deformation objects and use the online deformable sample creation module to create various hard deformation object samples. Note: During the pretraining process of ODEM-GAN, all parameters of the CBAF object detection model are frozen and not updated;
- Minimize Formulas (6) and (8) to alternately train the discriminator and the generator (note: update the discriminator parameters first, then update the generator parameters) until the deformable samples generated by the generator are enough to be real. Throughout the pretraining process, ODEM-GAN is trained for 30 epochs using a dataset containing only specific hard deformation objects.

  The purpose of pretraining is to allow the ODEM-GAN module to have a preliminary understanding of specific hard deformation object samples and reduce the difficulty of joint training of the ODEM-GAN module and CBAF object detection model.

**Joint pretraining:**

- The training set of joint training includes two parts: a dataset that only contains specific hard deformation object samples and a dataset that does not contain specific hard deformation object samples;
- Unfreeze all parameters of the CBAF object detection model and update the parameters;
- In each iteration, two images were taken from the datasets containing only specific hard deformation object samples and those without as the input of the network, so batch size = 4;
- After using the detector to extract the feature patches of the image containing specific hard deformation object samples and fuse the context information, use the training method of the pretraining process to update the parameters of the ODEM-GAN;
- Use the generator of ODEM-GAN to generate deformable samples of specific objects;
- Use the detector to extract feature patches that do not contain specific hard deformation object images and fuse context information;
- Place the fused feature patches of the specific hard deformation object images extracted by the detector, the fused contextual feature patches that do not contain the specific hard deformation object images, and the specific hard deformation object samples generated by ODEM-GAN together as a batch. Then, they are taken together as input to the classification and regression modules, and the optimizer is directly used to update the network parameters of the CBAF object detection model. Note that if the other specific category is changed, the GAN branch must be retrained.

**Inference:** The GAN branch of the ODEM-GAN module only participates in training and does not participate in prediction. Its prediction process is the same as that of the CBAF object detection model. Therefore, there is no additional calculation during the prediction, and the running speed of the detection algorithm will not be affected.

We only decoded box predictions whose confidence score threshold is higher than 0.05 and only selected up to at most 1k top-scoring in each pyramid level. The box predictions

from each pyramid level based on the anchor boxes branch and the anchor-free box branch are merged together; then, non-maximum suppression is performed with a threshold of 0.5 to produce the final detection result.

**Initialization:** We use ImageNet1k [33] to pre-train the backbone network. For convolutional layers in our CBAF module, the classification and regression layers are initialized as in [13]. For the initialization of the convolutional layer and the fully connected layer in the generator and discriminator, this paper uses "Xavier" initialization [34].

**Optimization:** The loss of the whole network consists of the original RetinaNet loss and the ODEM-GAN loss, as shown in Equation (12):

$$L = L_{ret} + \lambda L_{ODEM-GAN}, \tag{12}$$

where $L_{ret}$ is the loss of RetinaNet and $\lambda$ controls the weight of the ODEM-GAN module. In this article, we set $\lambda = 0.5$. A stochastic gradient descent (SGD) optimizer is used to train the whole network. The entire network is trained on two GPUs, and the batch size is four. For the CBAF module, the initial learning rate of training is 0.01, which is divided by 10 at 10 K, 60 K and 90 K iterations. For the GAN branch, we set the weight decay to 0.0005 and the momentum to 0.9. For both the generator and the discriminator, the learning rate is set to 0.00008, and the learning rate decays dynamically, decaying to 0.00002 during 50 K iterations. The batch size is set to four.

## 4. Experiments

We experiment with the ODEM-GAN module on the MS-COCO dataset [2], our airdrop experiment dataset and VOC 2012. The training date of MS-COCO is 'trainval35k', which includes 80 K images from the train and 35 K images from the 40 k val. The airdrop dataset is made up of two parts: (1) the matching image that was scraped from the Internet using a web crawler approach; (2) relevant images that were taken during the actual airdrop experiments. Human faces, quadrotor UAVs, fixed-wing aircraft, and parachutes are among the categories in the airdrop dataset. Faces and quadrotor UAVs are among the objects introduced to help with system debugging because commissioning a system with an air-drop experiment is inefficient and expensive. Faces and quadrotor drones, which are ubiquitous targets in life, may be used to decrease costs and make debugging the system more convenient and adaptable. Take 9/10 of our dataset as the training set (20,373 images), and the remaining 1/10 as the test set (2264 images). Part of the training dataset (we cannot release the images collected during the airdrop experiments without the permission of the project funding provider, but we can release the images from the Internet) is available from https://pan.baidu.com/s/1nsZAWnVy-3rrVMZypNdnGg (accessed on 25 March 2022). We evaluated our module by ablation studies on our dataset. The experimental comparison with state-of-the-art methods was carried out on "test-dev" and our dataset, and the relevant AP value was calculated. To easily show the performance of each category, we calculated the AP value of each category on VOC 2012.

### 4.1. Ablation Studies

We used our airdrop experiment dataset in the ablation studies. In the airdrop experiment, the parachute had severe deformation, so we set the parachute as the specific hard deformation object. The resolution of the image is scaled to $800 \times 800$ for training and testing in ablation experiments, and we use ResNet-50 as the backbone network and all networks based on the CBAF module [22]. We mainly evaluated the impact of adding a conditional generator and the positioning accuracy loss on the detector.

When using the current feature patch as the condition of the generator and the generator does not add the condition, after using ODEM-GAN to enhance the training of the CBAF detection model, the CBAF detection model only has an AP score of 76.4% for parachute detection on the airdrop dataset (third row in Table 4), which is lower than the result of augmentation training without ODEM-GAN (second row in Table 4). When conditions are added to the generator, the AP score after enhanced training for the CBAF object detection

model is 85.1% (the fourth row in Table 4), an increase of 0.3%. The experimental results show that the current feature patch must be used as the input of the generator to improve the parachute detection performance of the CBAF object detection model. The reason for this situation is analyzed in this paper: if conditions are not added to the generator, the hard samples generated by ODEM-GAN have no directionality, which will interfere with the learning of the classification and regression modules, resulting in a decrease in accuracy. In contrast, after the generator adds conditions, the generated hard deformation parachute samples are under specific conditions, have clear directionality, and can enhance the detection ability of the classification and regression modules.

When the loss of positioning accuracy is added, after using ODEM-GAN to enhance the training of the CBAF model, the AP score of the CBAF model for parachute detection on the airdrop dataset is 88.4% (the fifth row in Table 4). The AP score increased by 3.6%. The experimental results show that when the generator adds the positioning accuracy loss, the generator will generate parachute deformation samples with clear position information and can effectively reduce the impact of bounding box jitter on the regression accuracy in consecutive frames. We use the intersection-over-union (IOU) of two adjacent frames as the evaluation criterion for bounding box jitter. If the size and position of the bounding boxes in the adjacent two frames of images change little, the overlap ratio of the bounding boxes in the two frames is high; otherwise, it is small. The experimental results of the bounding box IOU in adjacent frames are shown in Figure 11a,b, respectively. The experimental results show that the IOU in adjacent frames of the model without ODEM-GAN is only approximately 0.2; after the ODEM-GAN module is added, the IOU in adjacent frames is increased to approximately 0.78, and the jitter is effectively suppressed. After adding the deformation enhancement model, the detection effect of the detector on the parachute in the open parachute state is visualized, as shown in Figure 12a,b. Therefore, using these samples to enhance the training of the regression module can further improve the accuracy of the detector on the object. The comprehensive experimental results show that using ODEM-GAN to enhance the training of the CBAF model can effectively improve the robustness of the CBAF model to deformation parachute detection.

**Table 4.** ODEM-GAN ablation experiment results.

| 1st | ODEM-GAN | Generator Addition Conditions | Loss of Positioning Accuracy | AP (%) |
|-----|----------|-------------------------------|------------------------------|--------|
| 2nd | no | - | - | 84.8 |
| 3rd | yes | no | no | 76.4 |
| 4th | yes | yes | no | 85.1 |
| 5th | yes | yes | yes | 88.4 |

The experimental results are shown in Figure 13. We found that there are both guided parachutes (the smaller parachute in Figure 13) and the main parachute which is not fully opened. The original RetinaNet can only detect the larger parachute (the first row); our proposed CBAF model is for the small and deformable guided parachute detection performance is not robust enough (the second row, the second column, the third column); and the ODEM-GAN model can detect the guide parachute and the main parachute well.

The data shown in Table 4 are the AP score of the parachute detection by the CBAF target detection model on the airdrop dataset after using ODEM-GAN to enhance the training of the CBAF target detection model. Note: The data shown in the second row in Table 4 are the AP score for parachute detection by the CBAF target detection model on the airdrop data.

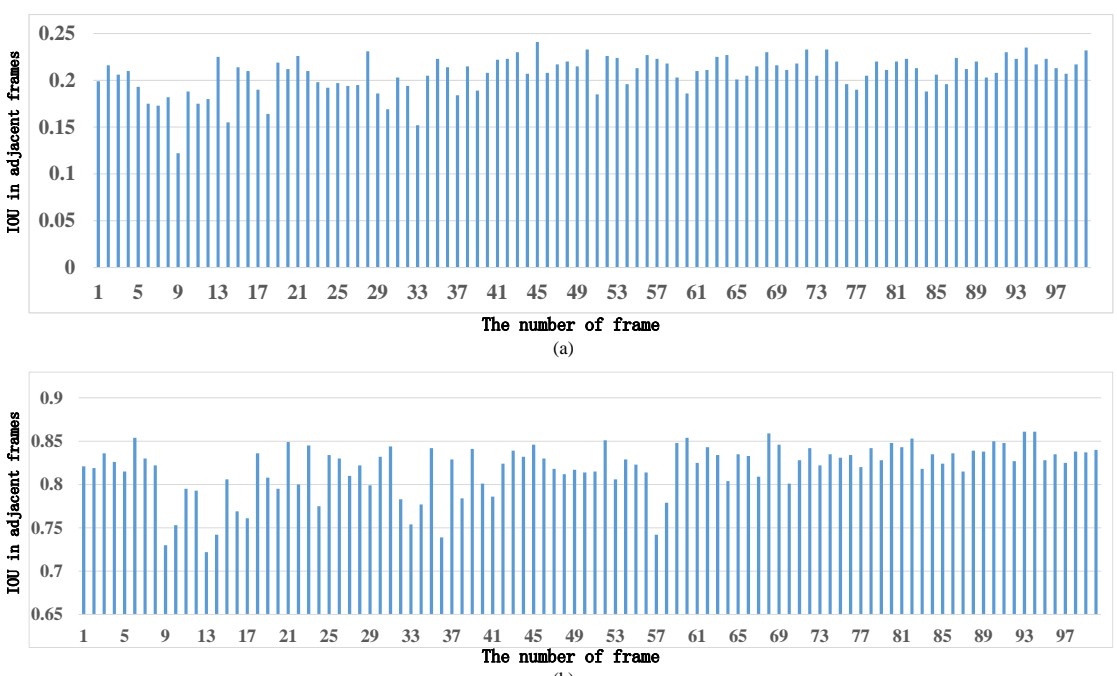

**Figure 11.** The IOU in adjacent frames: (**a**) the result of the CBAF module is only approximately 0.2; (**b**) the result of the ODEM-GAN module is increased to approximately 0.78, and the jitter is effectively suppressed.

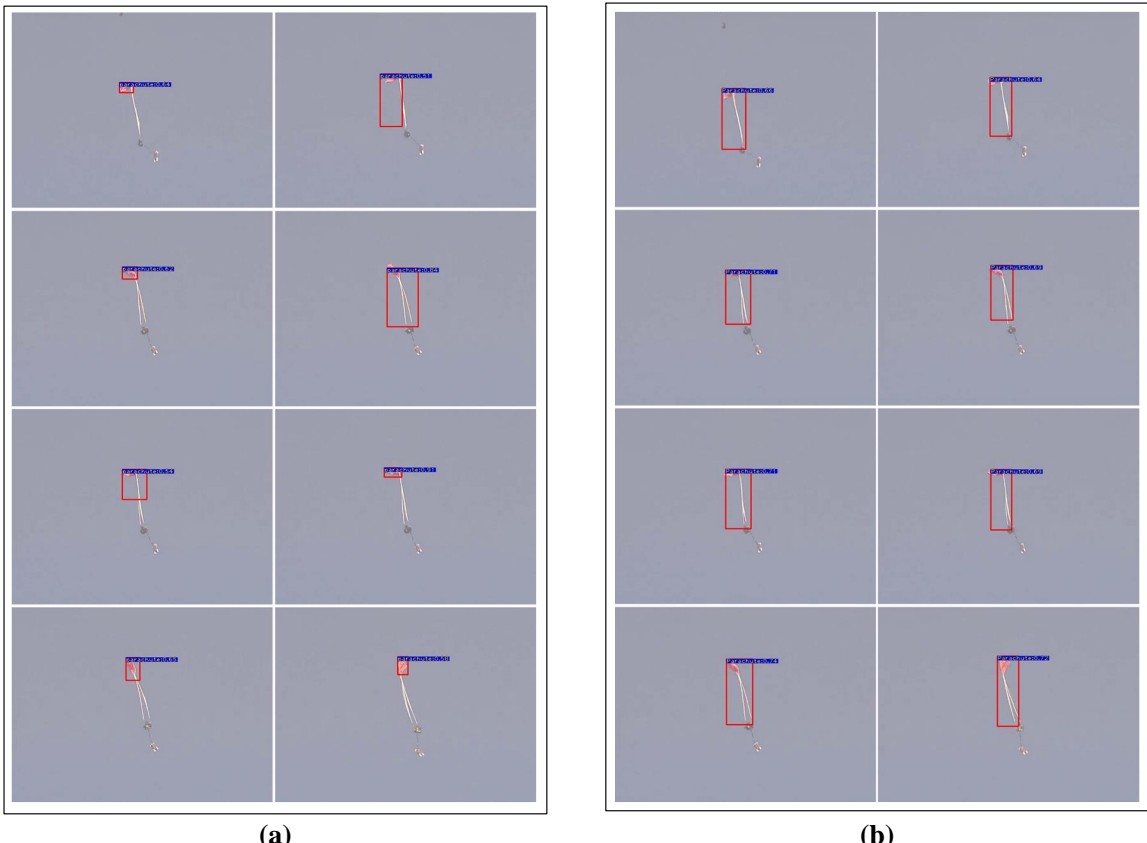

(**a**)  (**b**)

**Figure 12.** Visualization of the IOU in adjacent frames result: (**a**) the RetinaNet with CBAF module; and (**b**) the RetinaNet with ODEM-GAN module.

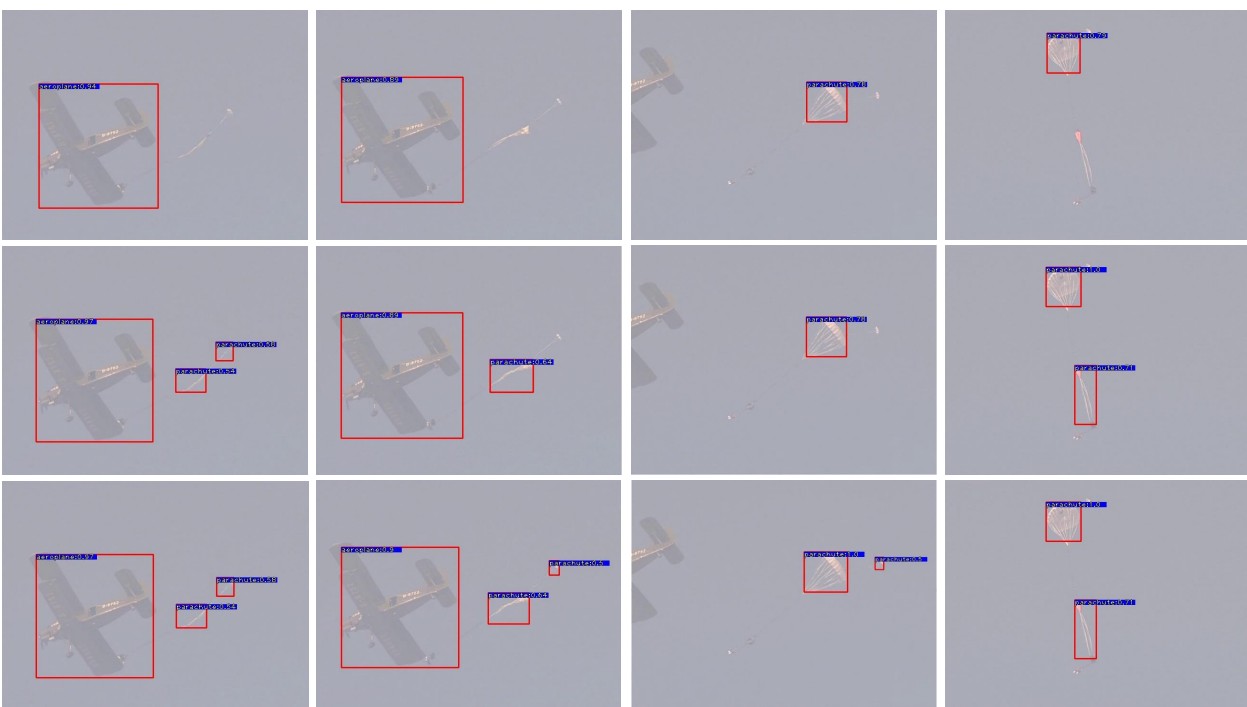

**Figure 13.** Comparison between our detector and the original RetinaNet. From the first row to the last row, the detection results of the original RetinaNet, the RetinaNet with the CBAF module, and the RetinaNet with the ODEM-GAN module are represented, respectively.

### 4.2. Determination of Hyperparameters

For the determination of the weight $\lambda_{lp}$ in Formula (8), we used different $\lambda_{lp}$ values to jointly train the CBAF module and ODEM-GAN module for training on our airdrop dataset, and the backbone is ResNet-50. The AP scores for parachute detection are shown in Tables 5 and 6. The experimental results show that when $\lambda_{lp} = 0.1$, the ODEM-GAN module achieves the best results for parachute detection.

**Table 5.** Experimental results using different $\lambda$ values.

| $\lambda$ | 0.1 | 0.2 | 0.3 | 0.4 | 0.5 | 0.6 | 0.7 | 0.8 | 0.9 |
|---|---|---|---|---|---|---|---|---|---|
| AP | 35.6 | 36.3 | 36.5 | 36.7 | 36.9 | 36.4 | 36.3 | 36.1 | 35.7 |

**Table 6.** Experimental results using different $\lambda_{lp}$ values.

| $\lambda_{lp}$ | 0.02 | 0.04 | 0.06 | 0.08 | 0.1 | 0.12 | 0.14 | 0.16 | 0.18 |
|---|---|---|---|---|---|---|---|---|---|
| AP (parachute) | 85.1 | 85.5 | 86.3 | 87.3 | 88.4 | 88.0 | 86.3 | 84.2 | 83.4 |

### 4.3. Results on PASCAL VOC 2012

A GAN can only generate one certain category of object. To easily show the performance of the ODEM-GAN module for each category, the PASCAL VOC 2012 dataset has 20 categories, and we optimized each category and calculated the AP value of each category on VOC 2012. We show the results of the original RetinaNet-50, the CBAF module, and our ODEM-GAN module on VOC 2012 in Table 7. We found that the AP score increased by 3% compared with ResNet-50, and there are varying degrees of improvement in AP scores for each category, which proves that our model can improve the performance of the detector very well.

**Table 7.** The performance based on the AP score of the ODEM-GAN module for each category on the PASCAL VOC 2012 dataset.

| Object | RetinaNet | CBAF | ODEM-GAN |
|--------|-----------|------|----------|
| mAP | 73.8 | 75.1 | 76.8 |
| aero | 85.3 | 86.8 | 87.5 |
| bike | 79.2 | 80.6 | 82.8 |
| bird | 76.9 | 78.2 | 79.5 |
| boat | 59.9 | 62.1 | 66.2 |
| bottle | 50.8 | 51.6 | 58.1 |
| bus | 80.9 | 82 | 83.2 |
| car | 73.3 | 75 | 77.1 |
| cat | 93.9 | 94 | 94 |
| chair | 56.5 | 59.1 | 60 |
| cow | 80.1 | 81 | 83 |
| table | 60.2 | 62.1 | 62.5 |
| dog | 89 | 89.9 | 89.9 |
| horse | 85.8 | 87 | 88.5 |
| mbike | 84.8 | 85.4 | 85.9 |
| person | 78.6 | 80 | 82.3 |
| plant | 45.6 | 47.2 | 50.1 |
| sheep | 71.3 | 72.3 | 74.3 |
| sofa | 70.6 | 71 | 75.6 |
| train | 82.6 | 83.2 | 83 |
| tv | 70.9 | 72.7 | 74 |

*4.4. Comparison to State of the Art*

We compare the ODEM-GAN module with other state-of-the-art detectors [32,32,35–50] on the test-dev split. Through the experimental results, we found that the detector based on ODEM-GAN achieved 27.9% in APS, which is 2% better than our previously proposed CBAF model, as shown in Table 8, indicating that our model not only enhances the deformation target detection, but also improves the performance of small object detection compared with state-of-the-art detectors [29,39,41,42,51]. We visualized some experimental results, as shown in Figure 14, and we performed both deformation and small object detection with good performance; in Table 8, we previously proposed CBAF with no improvement in detecting larger objects, which is because the global features of large objects are destroyed due to the feature maps divide to feature patches in the feature map division layer, but on the ODEM-GAN model. There is a 1.5% improvement, which is due to the invariance of the samples generated by the GAN branch learned by the detector.

In addition, DETR [41], DINO [43], GLIP [44], and DyHead [45] in Table 8 are all transformer-based object detection models, in which DINO and DyHead are based on the Swin-Transformer, although these algorithms implement higher general detection accuracy, because the transformer encoding method has a larger receptive field, which has a similar idea to our feature map division layer but needs to consume more training samples and consume more computing resources, resulting in lower operating rates. Although the QueryInst [46] algorithm has excellent performance in detection tasks, it is mainly used for instance segmentation, which is a query-based instance segmentation method. YOLOR-D6 [47] proposed a method to use semantic information in upper layers of CNNs to represent implicit knowledge and combine multiple discriminators for object detection, which has excellent performance and real-time performance. Comparing YOLOR-D6 and DINO, YOLOR-D6 makes full use of implicit knowledge, and DINO uses the transformer as a backbone, and its AP scores have been greatly improved, which is an improvement for ODEM-GAN as it provides a new direction for the future.

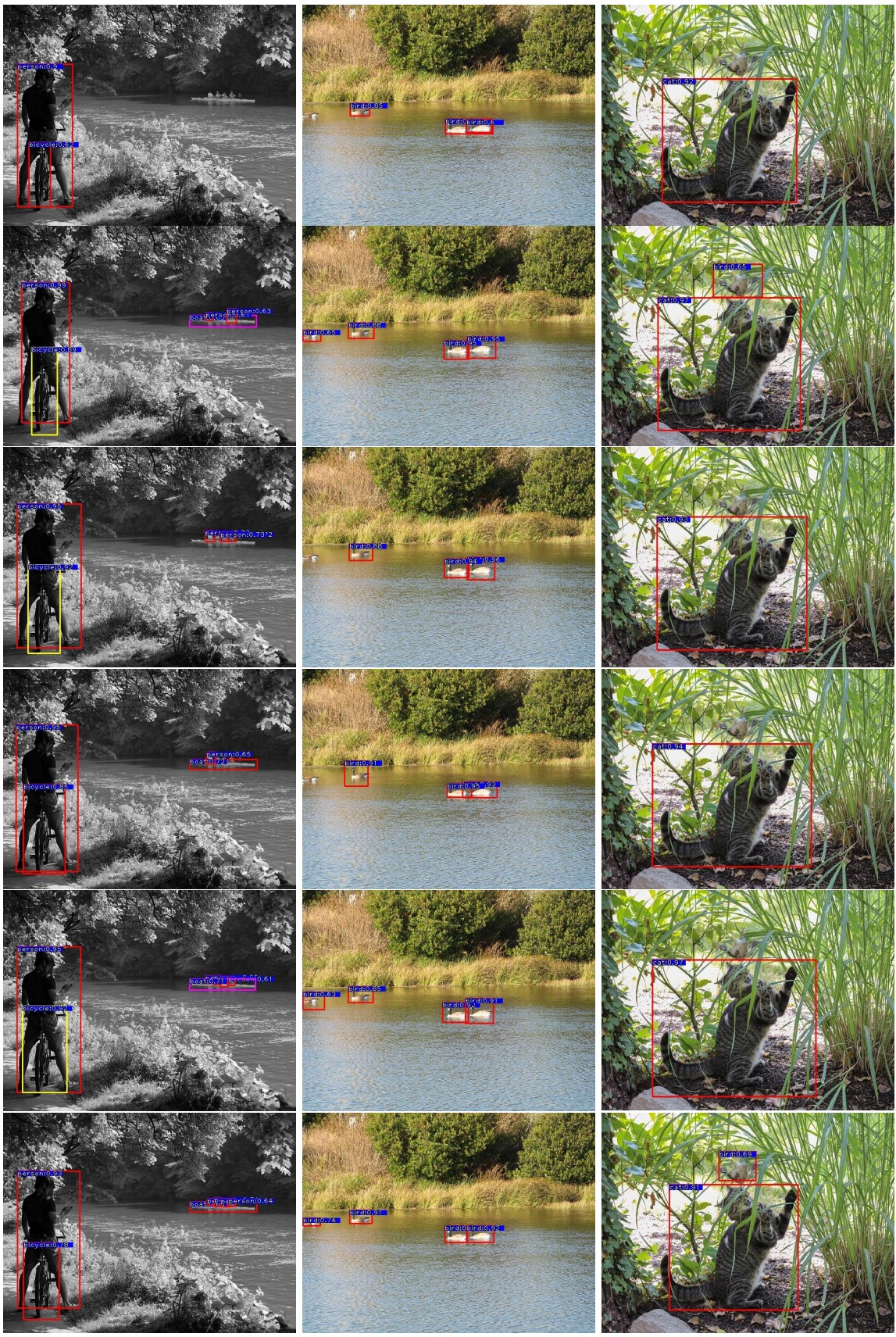

**Figure 14.** Comparison between our improved detector and recent state-of-the-art detectors. From the first row to the last row, the detection results of YOLOv4, EfficientDet-2, DETR, RDSNet, RetinaNet and ODEM-GAN, respectively.

**Table 8.** Results of the comparison of our improved detector (RetinaNet with our ODEM-GAN module) and state-of-the-art detectors.

| Methods | Backbone | $AP$ | $AP_{50}$ | $AP_{75}$ | $AP_S$ | $AP_M$ | $AP_L$ |
|---|---|---|---|---|---|---|---|
| DINO [43] | ResNet-50 | 50.2 | 68 | 54.7 | 32.8 | 53 | 64.8 |
| YOLOR-D6 [47] | YOLOv4-P6-light | 52.5 | 70.5 | 57.6 | 37.1 | 57.2 | 65.4 |
| YOLOv4 [35] | CSPDarknet-53 | 41.2 | 62.8 | 44.3 | 20.4 | 44.4 | 56 |
| CPN [38] | DLA-34 | 41.7 | 58.9 | 44.9 | 20.2 | 44.1 | 56.4 |
| EfficientDet-D2 [39] | Efficient-B2 | 43 | 62.3 | 46.2 | 22.5 | 47 | 58.4 |
| DETR [41] | Transformer | 43.5 | 63.8 | 46.4 | 21.9 | 48 | 61.8 |
| SM-NAS: E2 [40] | - | 40 | 58.2 | 43.4 | 21.1 | 42.4 | 51.7 |
| CenterNet [49] | Hourglass-104 | 44.9 | 62.4 | 48.1 | 25.6 | 47.4 | 57.4 |
| DINO [43] | Swin-T | 63.3 | 80.8 | 69.9 | 46.7 | 66 | 76.5 |
| DyHead [45] | Swin-T | 49.7 | 68 | 54.3 | 33.3 | 54.2 | 64.2 |
| GLIP [44] | Swin-T | 61.5 | 79.5 | 67.7 | 45.3 | 64.9 | 75 |
| QueryInst [46] | Swin-T | 49.1 | 74.2 | 53.8 | 31.5 | 51.8 | 63.2 |
| RDSNet [42] | ResNet-101 | 38.1 | 58.5 | 40.8 | 21.2 | 41.5 | 48.2 |
| CBAF [22] | ResNet-101 | 40.9 | 60.2 | 43.2 | 22.5 | 43.3 | 50.4 |
| DyHead [45] | ResNet-101 | 46.5 | 64.5 | 50.7 | 28.3 | 50.3 | 57.5 |
| QueryInst [46] | ResNet-101 | 42.8 | 65.6 | 46.7 | 24.6 | 45 | 55.5 |
| our (RetinaNet + ODEM-GAN) | ResNet-101 | 41.3 | 60.5 | 43.2 | 24.8 | 43.3 | 51.4 |
| Grid R-CNN w/FPN [48] | ResNeXt-101 | 43.2 | 63 | 46.6 | 25.1 | 46.5 | 55.2 |
| Dense RepPoints [37] | ResNeXt-101 | 40.2 | 63.8 | 43.1 | 23.1 | 43.6 | 52 |
| FoveaBox [50] | ResNeXt-101 | 42.1 | 61.9 | 45.2 | 24.9 | 46.8 | 55.6 |
| RetinaNet [7] | ResNeXt-101 | 40.8 | 61.1 | 44.1 | 24.1 | 44.2 | 51.2 |
| CBAF [22] | ResNeXt-101 | 43 | 63.2 | 46.3 | 25.9 | 45.6 | 51.4 |
| DyHead [45] | ResNeXt-101 | 47.7 | 65.7 | 51.9 | 31.5 | 51.7 | 60.7 |
| QueryInst [46] | ResNeXt-101 | 44.6 | 68.1 | 48.7 | 26.6 | 46.9 | 57.7 |
| our (RetinaNet + ODEM-GAN) | ResNeXt-101 | 43.2 | 63.4 | 46.8 | 27.9 | 45.6 | 52.9 |

## 5. Conclusions

In the continuous image sequence, the CBAF detection model will have the problem of bounding box jitter in the continuous detection of the object during the deformation process. This paper proposes ODEM-GAN to enhance the training of the CBAF detection model and improve the invariance of deformation for object detection. ODEM-GAN consists of three parts: an online deformable sample creation module, a generator, and a discriminator. The online deformable sample creation module is mainly used to create object deformation samples, the generator is used to generate diverse object deformation samples, and the discriminator is used to compete with the generator to ensure that the samples generated by the generator are sufficiently fake. In addition, based on the adversarial loss function of GAN, a positioning accuracy loss function is proposed, which effectively improves the regression accuracy of the detector. Through comprehensive comparative experiments, it is verified that the proposed ODEM-GAN can enhance the pose and deformation irrelevance of the CBAF detection model for parachute detection and effectively suppress the phenomenon of bounding box jitter.

Although our algorithm has obvious advantages for specific tasks, it has certain limitations in the training process, the training complexity is relatively high, and if the specific optimized objects are changed in the training of the process, it needs to be retrained, which is a limitation caused by the fact that GAN networks can only generate one category of specific objects. At the same time, because GAN branches are added to each level of the feature pyramid resulting in a large number of parameters, the training time will be relatively long. To address the above limitations, our future improvement can be based on two aspects: (1) improving the GAN network so that it can optimize multiple categories' objects to improve the robustness of the classifier; and (2) optimizing the network structure so that the output of each layer of the feature pyramid is connected to a unified GAN branch to reduce the complexity of training and shorten the training time.

**Author Contributions:** Conceptualization, Z.Z. and Z.P.; data collection, Z.T.; analysis and interpretation of results, Z.Z. and F.G.; validation, Z.Z. and F.G.; writing—original draft preparation, Z.Z. and F.G.; writing—review and editing, F.G. All authors have read and agreed to the published version of the manuscript.

**Funding:** This work was supported by the China Postdoctoral Science Foundation (2020M671597), Jiangsu Postdoctoral Research Foundation (2020Z100), the National Science Foundation of the Jiangsu Higher Education Institutions of China (20KJB520002), Suzhou Planning Project of Science and Technology (No. SYG202024), and the Priority Academic Program Development of Jiangsu Higher Education Institutions (PAPD).

**Institutional Review Board Statement:** Not applicable.

**Informed Consent Statement:** Not applicable.

**Data Availability Statement:** Not applicable.

**Conflicts of Interest:** The authors declare no conflict of interest.

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
