# Peer review of "ODEM-GAN: An Object Deformation Enhancement Model Based on Generative Adversarial Networks"

_applsci, doi:10.3390/app12094609_

Round 1

Reviewer 1 Report

Dear Authors,

Paper summary: This paper introduces a new method for small object detection using an object deformation enhancement model based on GAN. The proposed method has been validated by simulation for a parachute opinion process, which consists of a small object with various deformations throughout the opening process. The AP score for this simulated model is 88.4%.

Comments for the authors:

(1) Soundness: Very High

- Authors have presented the related work, ODEM-based GAN proposed method, and test results on different datasets very well. The authors have used 22637 images for the airdrop experiment. However, the authors should fix the following issues as listed.

Line 60: Wrong placement of period (cannot detect the parachute. As shown in Figure 1.)

Table 6: Improve visualization

Table 6: The title starts with “the”. Please correct.

Table 7: It is placed in between the conclusion. Please move “Table 7” before the conclusion.

For optimization, which hyperparameters are used and how hyperparameters are optimized need to explain more clearly.

(2) Significance: Medium

- The proposed concept is very good and can be applied in real life for example “Medical imaging for cancerous lesion detection using AI”. Cancerous lesions are very small in size and change in size for certain diseases like micro-calcification of the dense breast. The author should think more deeply, about how this approach will be applied in real life. If this manuscript cannot be improved at present with the real life application then add some real-life applications as future work.

(3) Novelty: High

- Authors have presented a new method for small object detection using an object deformation enhancement model based on GAN with improved accuracy. This can be applied in real life.

(4) Scientific Writing: Very Good

- Authors’ manuscript is very well writeen.

Author Response

Comment 1.1

(1) Soundness: Very High

Response 1.1

Many thanks for your positive comments.

Comment 1.2

Line 60: Wrong placement of period (cannot detect the parachute. As shown in Figure 1.)

Response 1.2

Many thanks for your valuable comments. We are sorry that we didn’t explain the process of parachute movement clearly. As shown in Figure 1(a to d), the movement process of the parachute is divided into four stages: 1. The free fall stage of the airdropped item leaving the aircraft(Figure 1a); 2. The stage of pulling out the main parachute from the umbrella bag(Figure 1b); 3. The stage of inflating the main parachute(Figure 1c); 4. The stage of stable descent of the main parachute(Figure 1d). If we choose the parachute in the stable descent phase as the training sample, it is difficult for the detector to detect the parachute in the first three phases. Because the parachute is gradually opened, its violent shape change will cause the confidence score during the detection process to drop sharply, resulting in detection failure (the detection object is lost multiple times during the experiment).

Comment 1.3

Table 6: Improve visualization

Response 1.3

Thanks for your comments. We have revised Table 6 in the manuscript which is not easy to view

Comment 1.4

Table 6: The title starts with “the”. Please correct.

Response 1.4

Thanks for your comments. We have revised the case error of the title of Table 6 in the manuscript.

Comment 1.5

Table 7: It is placed in between the conclusion. Please move “Table 7” before the conclusion.

Response 1.5

Thanks for your comments. We have moved the position of Table 7 in the manuscript and placed it before the conclusion.

Comment 1.6

For optimization, which hyperparameters are used and how hyperparameters are optimized need to explain more clearly.

Response 1.6

Thanks for your comments. We are sorry that we didn’t explain the hyperparameters of parachute movement clearly.  is the strength of ODEM-GAN module to control the loss contribution to the overall loss. we set . To determine the value of , we have done a series of experiments with different values of  on the minival, using ResNet-50 as the backbone. The AP scores for parachute detection are shown in Table 1. And the experimental results show that when  = 0.5, the ODEM-GAN module achieves the best results. The batch size is set to 4 For the CBAF module, the initial learning rate of training is 0.01, which is divided by 10 at 10K, 60K and 90K iterations. For the GAN branch, we set the weight decay to 0.0005 and the momentum to 0.9. For both the generator and the discriminator, the learning rate is set to 0.00008, and the learning rate decays dynamically, decaying to 0.00002 during 50K iterations.

Table 1. Experimental results using different  values

0.1

0.2

0.3

0.4

0.5

0.6

0.7

0.8

0.9

AP

35.6

36.3

36.5

36.7

36.9

36.4

36.3

36.1

35.7

For the determination of the weight in formula (8), we use different values ​​to jointly train the CBAF module and ODEM-GAN module for training on our airdrop data set, and the backbone is ResNet-50. The AP scores for parachute detection are shown in Table 5. The experimental results show that when = 0.1, the ODEM-GAN module achieves the best results for parachute detection.

Table 5. Experimental results using different values

0.02

0.04

0.06

0.08

0.1

0.12

0.14

0.16

0.18

AP(parachute)

85.1

85.5

86.3

87.3

88.4

88.0

86.3

84.2

83.4

Comment 1.7

(2) Significance: Medium

- The proposed concept is very good and can be applied in real life for example “Medical imaging for cancerous lesion detection using AI”. Cancerous lesions are very small in size and change in size for certain diseases like micro-calcification of the dense breast. The author should think more deeply, about how this approach will be applied in real life. If this manuscript cannot be improved at present with the real life application then add some real-life applications as future work.

Response 1.7

Many thanks for your valuable comments. We strongly agree with your suggestion and also we are now using this method in histological images of kidney to detect kidney cancer and MRI to determine the detection of suggestive Alzheimer's disease, both of which need to be identified on small size features, and also you have inspired us to apply it in dense breast microcalcifications. In the future we need to continue to improve this method in an effort to achieve better detection in the medical field and to provide a general idea for detection and segmentation tasks in the medical image field. At the same time, our method needs to add GAN branches on each level of the pyramid features, which is a process that consumes computational resources, so we need to make some improvements and optimization in the computational resource consumption.

Comment 1.8

(3) Novelty: High

- Authors have presented a new method for small object detection using an object deformation enhancement model based on GAN with improved accuracy. This can be applied in real life.

Response 1.8

Many thanks for your positive comments.

Comment 1.9

(4) Scientific Writing: Very Good

- Authors’ manuscript is very well writeen.

Response 1.9

Many thanks for your positive comments.

Reviewer 2 Report

  • Please include an introductory paragraph before subsection 2.1, explaining how the relevant literature for this study was obtained. Were the authors based on any state of the art of their own or of other authors? Was it complemented with the criteria of a panel of experts? It is important to let readers know why the works cited are necessary and sufficient as a starting point for the proposal.
  • Although this is not a main concern, try to place the figures and tables as close as possible to the prose.
  • Include at the end of section 2 a paragraph that summarizes the shortcomings detected in the reference methods used as a basis for comparison for your proposal.
  • When datasets are used for experimentation, it is good practice to indicate in advance why those datasets are appropriate for the research objectives. Are they standard within the community that researches the topic? Are they the only ones that exist? Are all the ones that the community uses regularly being used, or is there a reason to omit some of them? On the other hand, it is necessary to clarify how your airdrop experiment dataset is made up. Please clarify this in section 4 Experiments.
  • I suggest expanding the Conclusions section with some ideas for future research.

Author Response

Comment 2.1

Please include an introductory paragraph before subsection 2.1, explaining how the relevant literature for this study was obtained.

Response 2.1

Thank you for your valuable comments. We add an introductory paragraph before subsection 2.1, the paragraphs are as follows:

Most object detection approaches are classified as one-stage methods and two-stage methods. The two stages approach for object detection includes two steps. First, extraction of multiple regions of interest (RoIs) using region proposal network. Second, these ROIs are classified and refined using classification and regression module, respectively. Unlike the two-stage method, the one-stage method predicts the item immediately using a single neural network structure. The one-stage approach has a faster prediction speed than the two-stage method, but its accuracy is not as excellent. Among them, YOLO, RetinaNet, CornerNet, etc. are the most representative methods, and the contributions of these methods are described in detail in the next paragraph. Feature pyramids, also known as multi-level feature towers, are progressively becoming a frequent structure for object detection, since it handle object scale variances and produce considerable results. SSD[1] first predicts class scores and bounding boxes at different resolutions of feature maps so that objects of different sizes can be processed. FPN [3] and DSSD [2] proposed to a top-down architecture with lateral connections, which enhance low-level features with high-level semantic feature maps at all scales. In order to better adapt to the scale changes of anchors and to solve the limitations of anchor boxes, the anchor-free approach has attracted much attention in recent years. Among them YOLO, CornerNet left a deep impression. CornerNet solved object detection as the key point detection problem, the prediction box is obtained by detecting two key points in the upper left corner and the lower right corner.

[1]  W. Liu, D. Anguelov, D. Erhan, C. Szegedy, S. Reed, C.Y. Fu, and A. C. Berg. Ssd: Single shot multibox detector. In European conference on computer vision, pages 21–37. Springer, 2016. 2, 3, 8

[2]  C.-Y. Fu, W. Liu, A. Ranga, A. Tyagi, and A. C. Berg. Dssd: Deconvolutional single shot detector. arXiv preprint arXiv:1701.06659, 2017. 2, 3, 8

[3]  T.-Y. Lin, P. Dollar, R. B. Girshick, K. He, B. Hariharan, and S. J. Belongie. Feature pyramid networks for object detection. In CVPR, page 3, 2017. 2, 5, 8

Comment 2.2

Were the authors based on any state of the art of their own or of other authors?

Response 2.2

    Thank you for your valuable comments. Our ODEM-GAN module based on state of the art of CBAF [22] module. CBAF [22] module is proposed by us last year. CBFA module chooses RetinaNet[13] as the baseline and used ResNet as the backbone.

Comment 2.3

Was it complemented with the criteria of a panel of experts? It is important to let readers know why the works cited are necessary and sufficient as a starting point for the proposal.

Response 2.3

Thank you for taking the time to leave your important feedback. Our strategy was supplemented by a panel of experts' criteria. Wang, X [29] et al. suggested a method for improving the stability of classifiers by using adversarial networks to generate masks. The approach works well for occlusions, but it is less successful for objects with a lot of deformation. The deformation objects follow a long-tailed distribution, making it impossible to get the object's deformation in all situations through data collection. The model discriminator, the generator, and the perceptual GAN [30] are addressed to identify the generated representation and put conditional constraints on the generator; the created object representation must be favorable to object detection. This approach demonstrates the viability of using GAN networks for object detection, as well as demonstrating that the generated samples help in improving the classifier's accuracy. As a result, the approaches provided by Wang, X [29], Li, J [30], and other have inspired us. To create deformation samples, we utilize GAN networks. Unlike Wang, X[29] Li, J[30] et al., we do not use masks or generate similar training samples in our approach. We create deformation samples in order to recognize objects that have undergone significant deformation. By dividing the feature map, Vit [23] yolo [12,17,24] achieves the formation of object contextual relationships, enhancing the accuracy of small object detection.

Comment 2.4

Although this is not a main concern, try to place the figures and tables as close as possible to the prose.

Response 2.4

Thanks for your comments. We have moved the position of Table 7 in the manuscript and placed it before the conclusion and revised Table 6 in the manuscript which is not easy to view.

Comment 2.5

Include at the end of section 2 a paragraph that summarizes the shortcomings detected in the reference methods used as a basis for comparison for your proposal.

Response 2.5

Thank you for your valuable comments. We add an summarizes paragraph before subsection 2, the paragraphs are as follows:

Therefore we are inspired by the methods proposed by Wang, X[29] Li, J[30] et al. We use GAN networks to generate deformation samples. Unlike the method proposed by Wang, X[29] Li, J[30] et al, our method does not use masks or generate similar training samples. We generate deformation samples to achieve detection of objects with dramatic deformation. For the problem that the accuracy of the one-stage method is not as good as that of the two-stage method, we use focal loss [13] to solve it. the Vit[23]yolo[12,17,24] algorithm achieves the establishment of object contextual relations by divide the feature map, thus improving the accuracy of small target detection. Vit[23] is a transformer-based detection network, which is disadvantageous in that it requires a large amount of training data, while Yolo [12,17,24] is not as accurate as the two-stage network, although its prediction is fast. Inspired by the algorithm of Vit [23] yolo [12,17,24], we also use object contextual relations to improve the accuracy of small object detection. Inspired by combining each of these algorithms, we propose the ODEM-GAN module.

。

Comment 2.6

When datasets are used for experimentation, it is good practice to indicate in advance why those datasets are appropriate for the research objectives.

Response 2.6

Thank you for your valuable comments. We experimented with the ODEM-GAN module on the MS-COCO dataset [2], the airdrop experiment dataset and VOC 2012. the training date of MS-COCO is ' trainval35k ', which includes 80K images from trains and 35K images from 40k val. Our purpose of using the MS-COCO dataset is to compare it with state of the art and to verify the common ability of our algorithm. The purpose of using the airdrop experiment dataset is to demonstrate that our algorithm can solve our recognition task in the airdrop experiment and to verify the effectiveness of our algorithm over the baseline network. Limited by the fact that GAN can only generate a specific category of objects, and to facilitate demonstrating the optimization performance of the ODEM-GAN module for each specific category, we used each category from the PASCAL VOC 2012 dataset, which has 20 categories, for optimization and calculated the AP values for each category on VOC 2012.

Comment 2.7

Are they standard within the community that researches the topic? Are they the only ones that exist? Are all the ones that the community uses regularly being used, or is there a reason to omit some of them?

Response 2.7

Thank you for your valuable comments. The AP scores are the standard that researches the topic. AP is the average of the algorithm's detection accuracy over all categories. Traditionally, mAP (mean average precision) was used as the average of the algorithm's detection accuracy for all categories, but nowadays AP is no longer distinguished from mAP. In order to be consistent with the way it has been written in recent years in published papers, AP is used in this paper. More specifically, MS-COCO gives six evaluation metrics, AP, AP50, AP75, APS, APM, and APL. AP is the value taken over multiple intersection and merge ratio (IOU=0.5:0.05:0.95) values. AP50 and AP75 are the accuracies at IOU=0.5 and 0.75, respectively. aps indicates the accuracy of detecting small objects (area less than 322, in pixels), APM indicates the accuracy of detecting medium-sized objects (area larger than 322 and smaller than 962), and APL indicates the accuracy of detecting large objects (area larger than 962). For more description of the evaluation metrics and how to use the evaluation code, visit the MS-COCO website: https://cocodataset.org/#detection-eval. Note: "test-dev" has no publicly available label and needs to be calculated using the MS-COCO evaluation server provided to calculate it.

We use inter-frame IOU [1] which is often found in tracking tasks. Since some of the samples in our dataset are image sequences, which are similar to the data of tracking tasks, we choose the inter-frame IOU commonly used in tracking tasks as the evaluation standard, other evaluation standards are not applicable to our detection tasks, so we do not choose

[1]  Bochinski E, Eiselein V, Sikora T. High-speed tracking-by-detection without using image information[C]//2017 14th IEEE international conference on advanced video and signal based surveillance (AVSS). IEEE, 2017: 1-6.

Comment 2.8

On the other hand, it is necessary to clarify how your airdrop experiment dataset is made up. Please clarify this in section 4 Experiments.

Response 2.8

Thank you for your valuable comments. The airdrop dataset is made up of two parts:1. matching image that was scraped from the Internet using a web crawler approach, 2. relevant images that were taken during the actual airdrop experiments. Human faces, quadrotor UAVs, fixed-wing aircraft, and parachutes are among the categories in the airdrop dataset. Faces and quadrotor UAVs are among the objects introduced to help with system debugging. Because commissioning a system with an air-drop experiments is inefficient and expensive. Faces and quadrotor drones, which are ubiquitous targets in life, may be used to decrease costs and make debugging the system more convenient and adaptable. Take 9/10 of the our data set as the training set (20373 images), and the remaining 1/10 as the test set (2264 images).

Comment 2.9

I suggest expanding the Conclusions section with some ideas for future research.

Response 2.9

Although our algorithm has obvious advantages for specific tasks, it has certain limitations in the training process, the training complexity is relatively high, and if the specific objects optimized are changed in the training of the process, it needs to be retrained, which is a limitation caused by the fact that GAN networks can only generate one category of specific objects. At the same time, because GAN branches are added to each level of the feature pyramid resulting in a large number of parameters, the training time will be relatively long. To address the above limitations, our future improvement can be based on two aspects: 1. improving the GAN network so that it can optimize multiple categories objects to improve the robustness of the classifier; 2. optimizing the network structure so that the output of each layer of the feature pyramid is connected to a unified GAN branch to reduce the complexity of training and shorten the training time.
